# Type 2 Deiodinase Thr92Ala Polymorphism Is Not Associated with Cognitive Impairment in Older Adults: A Cross-Sectional Study

**DOI:** 10.3390/metabo12050375

**Published:** 2022-04-21

**Authors:** Wallace Klein Schwengber, Vitor Bock Silveira, Guilherme Moreira Hetzel, Amanda Robaina, Lucieli Ceolin, Marli Teresinha Camelier, Iuri Goemann, Roberta Rigo Dalla Corte, Rafael Selbach Scheffel, Renato Gorga Bandeira de Mello, Ana Luiza Maia, José Miguel Dora

**Affiliations:** 1Thyroid Unit, Hospital de Clínicas de Porto Alegre, Medical School, Universidade Federal do Rio Grande do Sul, Porto Alegre 90035-003, Brazil; wschwengber@hcpa.edu.br (W.K.S.); vbsilveira@hcpa.edu.br (V.B.S.); ghetzel@hcpa.edu.br (G.M.H.); arobaina@hcpa.edu.br (A.R.); lu.ceolin@gmail.com (L.C.); mcamelier@hcpa.edu.br (M.T.C.); igoemann@gmail.com (I.G.); rscheffel@hcpa.edu.br (R.S.S.); amaia@hcpa.edu.br (A.L.M.); 2Geriatric Unit, Internal Medicine Division, Hospital de Clínicas de Porto Alegre, Medical School, Universidade Federal do Rio Grande do Sul, Porto Alegre 90035-003, Brazil; rrcorte@hcpa.edu.br (R.R.D.C.); rgmello@hcpa.edu.br (R.G.B.d.M.); 3Department of Pharmacology, Instituto de Ciências Básicas da Saúde, Universidade Federal do Rio Grande do Sul, Porto Alegre 90035-003, Brazil

**Keywords:** Type 2 deiodinase, DIO2 Thr92Ala polymorphism, iodothyronine deiodinase Type II, Thr92Ala polymorphism, cognition, older adults

## Abstract

Background: Type 2 Deiodinase (DIO2) converts thyroxine (T4) into the active hormone triiodothyronine (T3). Thr92Ala DIO2 polymorphism has been associated with reduced conversion of T4 into T3 and central nervous system hypothyroidism. However, how Thr92Ala DIO2 polymorphism affects cognitive function is still unclear. Objective: To assess the association between Thr92Ala DIO2 polymorphism and cognitive performance in older adults. Design: Cross-sectional study. Setting: University-based tertiary hospital in Brazil. Patients: > 65-year-old with no limiting clinical disease. Interventions: All participants answered a standard questionnaire before undergoing thyroid function laboratory evaluation and genotyping of the Thr92Ala DIO2 polymorphism. Main Outcomes: Cognitive impairment measured by the Word List Memory task from the Consortium to Establish a Registry for Alzheimer’s Disease Neuropsychological Battery (CERAD-NB) and the Brief Cognitive Screening Battery (BCSB). Results: A hundred individuals were included. Clinical and laboratory characteristics were similar among DIO2 genotypes (all *p* > 0.05). No differences were found in the Word List Memory, recall, or recognition tests of the CERAD-NB assuming a recessive model for the Ala/Ala vs. Thr/Ala-Thr/Thr genotypes. Results of Clock Drawing Test, Animal Fluency Test, Mini-Mental State Exam, and Figure Memory Test of the BCSB were similar between groups. Conclusions: These findings suggest that Thr92Ala DIO2 polymorphism is not associated with relevant cognitive impairment in older adults.

## 1. Introduction

Deiodinases transforms the prohormone thyroxine (T4) into the biologically active triiodothyronine (T3) by peripheral conversion. The Type 2 deiodinase (DIO2) enzyme is essential to maintain adequate levels of T3 in the intracellular environment of specialized tissues, such as the central nervous system (CNS) [1]. The Thr92Ala (rs225014) polymorphism in the Type 2 deiodinase (DIO2) gene has been associated with a decreased conversion of T4 to T3 [2,3] and several clinical [2,3,4,5,6] and cognitive syndromes [7,8,9].

Thyroid hormones are essential to develop and maintain cognition throughout life [10]. Hypothyroidism deteriorates cognitive function at any age [11]. Adults with subclinical hypothyroidism underperformed in cognitive tests when compared to euthyroid subjects in previous series [12]. Experimental data demonstrated that Thr92Ala DIO2 polymorphism is associated with decreased conversion of T4 into T3 in the CNS [13], intracellular oxidative stress, and neurodegeneration independent of T3 signaling [14].

The association between Thr92Ala DIO2 polymorphism and cognitive impairment in clinical studies is controversial [7,8,9,15,16,17,18]. The assessment of early cognitive impairment in older adults using validated questionnaires allows for evaluating several cognitive domains, including language, orientation, memory, concentration, praxis, learning, and executive function [19,20,21]. Wouters et al. [18] analyzed nonverbal fluency and executive functioning in a large cohort of patients, finding no association between Thr92Ala DIO2 polymorphism and cognitive functioning. Therefore, it is still unclear how Thr92Ala DIO2 polymorphism affects cognition in older adults. Given the prevalence of the 92Ala DIO2 allele of nearly 40% in different populations, clarifying the potential impact of the polymorphism on cognition is of special interest.

This study assessed the hypothesis that individuals > 65 years old homozygous for the Ala/Ala DIO2 genotype underperform in cognitive tests compared to the control group.

## 2. Results

### 2.1. Patients

Based on the selection criteria, 100 participants were enrolled in this study. The mean age was 72 ± 6 years old and 61 participants (61%) were women. The median serum TSH was 1.8 mU/L (IQR 1.3–2.6) and the median FT4 was 1.2 ng/dL (IQR 1.06–1.29). Table 1 shows clinical and laboratory characteristics of the population.

Regarding the DIO2 genotype distribution, 37 (37%) individuals were homozygous for the Thr allele, 45 (45%) were heterozygous (Thr/Ala), and 19 (19%) were homozygous for the Ala allele. The frequency of the minor Ala allele was 0.41 and the genotypes were in Hardy–Weinberg equilibrium (*p* = 0.94). Clinical and laboratory baseline characteristics were similar among DIO2 genotypes (Table 1).

### 2.2. Cognitive Tests

#### 2.2.1. Immediate/Delayed Memory and Learning

Assuming a recessive model for the Ala/Ala vs. Thr/Ala-Thr/Thr genotypes, respectively, no differences were found in the CERAD Word List memory (mean, 15.8 [95% confidence interval {95% CI}, 13.8–17.8] vs. 15.8 [95% CI 14.9–16.7], *p* = 1.0), recall (4.4 [95% CI 3.6–5.2] vs. 4.2 [95% CI 3.8–4.7], *p* = 0.7), and recognition tests (7.4 [95% CI 6.3–8.5] vs. 8.3 [95% CI 7.8–8.7], *p* = 0.1).

#### 2.2.2. Executive Function and Visuospatial Ability

No difference was observed in the Clock Drawing Test (7.3 [95% CI 6.0–8.6] vs. 7.8 [95% CI 7.2–8.3], *p* = 0.4).

#### 2.2.3. Language

Results were equal for verbal fluency (13.8 [95% CI 12.3–15.3] vs. 13.9 [95% CI 13.1–14.6], *p* = 0.7).

#### 2.2.4. Visual Perception and Nomination

The Figure Memory Test for visual perception (9.8 [95% CI 9.7–10.0] vs. 9.9 [95% CI 9.8–10.0], *p =* 0.6), nomination (9.9 [95% CI 9.8–10.0] vs. 9.9 [95% CI 9.8–10.0], *p* = 0.5), incident recall (6.5 [95% CI 6.0–7.0] vs. 6.8 [95% CI 6.5–7.1], *p =* 0.4), immediate recall 1 (8.0 [95% CI 7.3–8.7] vs. 7.8 [95% CI 7.4–8.2], *p* = 0.7), and immediate recall 2 (8.2 [95% CI 7.4–8.9] vs. 8.3 [95% CI 7.9–8.6], *p* = 0.8) did not differ between groups.

#### 2.2.5. Orientation, Concentration, and Praxis

Lastly, MMSE results (24.6 [95% CI 22.8–26.4] vs. 23.7 [95% CI 22.9–24.6], *p* = 0.4) were equal between groups (Figure 1).

## 3. Discussion

In this study, euthyroid older adults with no previous cognitive impairment answered an individualized comprehensive questionnaire to assess various cognitive domains, including memory, learning, executive function, visuospatial ability, language, visual perception, nomination, orientation, concentration, and praxis. The scores demonstrate that Ala/Ala and Thr/Ala-Thr/Thr DIO2 genotype groups were equal in all subtests, reinforcing that DIO2 Thr92Ala polymorphism has no clinically relevant association with any of the cognitive dimensions evaluated. Studied individuals and controls of previous studies [19,20,21] had similar scores in the Word List Memory Task of CERAD-NB and BCSC, showing that the evaluated population has a healthy cognition.

Reduced circulating thyroid hormones modify cerebral blood flow and neurotransmission, affecting mental activities related to the acquisition, storage, retrieval, and use of information (cognition) [22,23,24,25]. Previous studies revealed that subclinical hypothyroidism is associated with underperformance in cognitive tests [12]. In human brains homozygous for Thr92Ala DIO polymorphism, experimental data exhibited intracellular modifications independent of thyroid hormone signaling [14], indicating that Thr92Ala DIO2 polymorphism can be a risk factor for neurocognitive symptoms in euthyroid individuals.

Previous data demonstrate that the Ala92Ala genotype of the Thr92Ala DIO2 polymorphism reduces enzyme activity, leading to a state of intracellular hypothyroidism [2,3]. Studies on the effects of Thr92Ala DIO2 polymorphism on cognitive function are still incipient and controversial (Table 2). Luo et al. [8] identified that men with cognitive deficits had higher Ala92 allele frequency than the controls (49% vs. 39%, *p =* 0.08). In another study with patients with treated hypothyroidism, Thr92Ala DIO2 polymorphism did not affect neurocognitive functioning [17]. Wouters et al. [18] explored the association between DIO2 Thr92Ala polymorphism and neurocognition along with thyroid function tests assessment in Subjects from the Lifelines Cohort study (*n* = 12,625, General population 48 ± 11 years-old, and LT4 users 53 ± 12 years-old), using the Ruff Figural Fluency Test. They concluded that the Thr92Ala polymorphism was not associated with cognitive functioning, neither in the general population nor in subjects on thyroid hormone replacement therapy or matched controls. We studied a nearly 30 years-older population (our subjects had a mean 72 ± 6 years-old), using multiple cognitive tests that encompassed different cognitive dimensions. Our findings regarding older adults, with different methodologies, corroborate with those of Wouters et al. [18], who found no association between Thr92Ala DIO2 polymorphism and cognitive function in a large cohort study. Altogether both studies sum to a better understanding of the lack of association of the Thr92Ala DIO2 polymorphism and cognition. 

Nevertheless, the data suggest that metabolic dysfunction associated with low thyroid hormones causes late neurofunctional compromise after years of exposure [23,24,25]. Our hypothesis might therefore have been negative for analyzing cognitively healthy older adults. Further testing it in a population of patients with mild cognitive impairment (MCI) could contribute with the data on cognitive performance. The prevalence of the Thr92Ala DIO2 genotype could also be compared between cognitively healthy participants with MCI and participants with Alzheimer’s disease. Lastly, longitudinal studies could contribute by comparing incidence rates of conversion to dementia between individuals with MCI alone and individuals with the MCI and Thr92Ala DIO2 genotype.

This is the first study on Thr92Ala DIO2 polymorphism solely designed to assess cognitive function in euthyroid subjects with no previous cognitive impairment. Although only 19 individuals of our sample size were homozygous for the Ala allele, the frequency of the minor Ala allele was identical to those reported in other groups of patients with the same genetic background, including healthy young patients [3] and diabetic patients [26]. Moreover, the sample size calculation estimated that the 100 included participants (19 Ala/Ala vs. 81 Thr/Thr-Thr/Ala) would be enough to detect a 2-point difference in the Mini-Mental Test with a 90% power. Most importantly, the narrow confidence intervals found suggest appropriate power to test the hypothesis.

Cognitive function is a complex neurobiological process with several interferences that are difficult to assess cross-sectionally. Nonetheless, we excluded individuals susceptible to suffering from interference in cognitive function because of disease or drug interaction. Moreover, cognitive function was objectively assessed using several validated tests.

Overall, our study suggests that Thr92Ala DIO2 polymorphism is not associated with clinically relevant cognitive impairment in older adults. Longitudinal studies aimed at detecting cognitive impairment over time could contribute to our results.

## 4. Materials and Methods

### 4.1. Patients and Study Design

Participants included were patients > 65 years old in the outpatient clinic of the Internal Medicine Division of Hospital de Clínicas de Porto Alegre (HCPA), a tertiary care university hospital in Southern Brazil. All patients that fulfilled the selection criteria were invited to participate. Exclusion criteria were: history of dementia or delirium; limiting psychiatric or neurological disease; drug addiction; use of antipsychotics, benzodiazepines, or anticholinergics; limiting hearing or visual illnesses; active malignant neoplasia; more than two hospitalizations in the past 12 months; chronic kidney disease stage V or in renal replacement therapy (hemodialysis or peritoneal dialysis); heart failure class III or IV; cirrhosis class C; current use of any drug that affects thyroid function (e.g., thioamides, levothyroxine, lithium, amiodarone); and history of thyroid dysfunction.

The information obtained from the study did not influence the patient’s diagnosis or treatment. The local Research Ethics Committee approved the study protocol (CAAE 55054816700005327/GPPG 2016-0358) and all patients signed an informed consent form.

### 4.2. Cognitive Function Assessment

All patients were submitted to the Word List Memory Task of the Consortium to Establish a Registry for Alzheimer’s Disease Neuropsychological Battery (CERAD-NB) (which includes Word List memory, recall, and recognition tests) [19] and to the Brief Cognitive Screening Battery (BCSB) (which includes the Clock Drawing Test, Verbal Fluency Test, Mini-Mental State Exam, and Figure Memory Test) [20,21].

These questionnaires allow for assessing distinct cognitive domains, including immediate/delayed memory and learning (Word List Memory Task), executive function and visuospatial ability (Clock Drawing Test), language (Verbal Fluency Test), visual perception, and nomination (Figure Memory Test). The Mini-Mental State Exam (MMSE) also evaluated orientation, concentration, and praxis. Scores were expressed as continuous variables and presented as means.

Participants were classified according to schooling level and family income. Firstly, the schooling level was subdivided into never went to school (0 years of schooling), primary (from 1 to 5 years of schooling), lower secondary (from 6 to 9 years of schooling), upper secondary (from 10 to 12 years of schooling), and/tertiary (over 12 years of schooling). Then, family income was assessed by the number of minimum wages (MW) (998 BRL; 1 USD = 5.38 BRL) and further subdivided into the following categories: less than 1 MW, from 1 to less than 2 MWs, from 2 to 5 MWs, and over 5 MWs.

### 4.3. Laboratory Tests

Serum samples for thyrotropin (TSH) and free thyroxine (FT4) were collected. TSH (normal range [NR] 0.35–4.94 mU/L) and free thyroxine (NR 0.70–1.48 ng/dL) were measured using a chemiluminescence immunoassay (CLIA, Abbott, Chicago, IL, USA). Inter-assay coefficients of variation were 2.6% for TSH and 3.8% for FT4. The reference ranges for laboratory values are TSH 0.35–4.94 mU/L and FT4 0.70–1.48 ng/dL.

### 4.4. Genotyping

DNA was extracted from peripheral blood leukocytes by a standardized procedure. Primers and probes contained in the Human Custom TaqMan Genotyping Assay 40x (Applied Biosystems, Foster City, CA, USA) were used for genotyping our samples. The predesigned TaqMan SNP Genotyping Assay^®^ C_15819951_10 was used to analyze the DIO2 gene (rs225014 [Thr92Ala]) SNP (Applied Biosystems, Foster City, CA, USA). One allelic probe was labeled with VIC dye and the other with FAM dye. The reactions were conducted in a 96-well plate with a total 5 µL reaction volume using 2 ng of genomic DNA, TaqMan Genotyping Master Mix 1x (Applied Biosystems, Waltham, MA, USA), and Custom TaqMan Genotyping Assay 1x. The plates were then positioned in a real-time PCR thermal cycler (7500 Fast Real PCR System; Applied Biosystems) and heated for 10 min at 95 °C, followed by 50 cycles of 95 °C for 15 s and 60 °C for 90 s. Fluorescence data files from each plate were analyzed using automated allele-calling software (SDS 2.1; Applied Biosystems).

Patients were classified as Ala/Ala, Thr/Ala, or Thr/Thr genotypes. All amplification reactions were performed twice. The genotyping success was over 95%, with a calculated error rate based on PCR duplicates of 0%.

### 4.5. Statistical Analyses

A sample of 98 patients was calculated based on an estimated absolute difference in the Mini-Mental State Examination score of two points between groups exposed (Ala/Ala) and not exposed (Thr/Thr-Thr/Ala) DIO2 genotypes, with a 90% power and 5% statistical significance level.

Results are expressed as frequencies, mean ± 95% confidence intervals, or median and interquartile range (IQR). To examine the main effect of the DIO2 Thr92Ala variant, the three genotypes (Thr/Thr, Thr/Ala, and Ala/Ala) were considered separately, then grouped into Thr/Thr-Thr/Ala and Ala/Ala groups. Allelic frequencies were determined by gene counting. Departures from the Hardy–Weinberg equilibrium were verified using χ^2^ tests. Clinical and laboratory data with normal distribution were compared using χ^2^ (categorical data) or Student’s *t*-test for independent groups (continuous variables). A two-tailed *p* < 0.05 was considered statistically significant, with a Bonferroni correction for multiple analysis. All analyses were performed with SPSS version 20.0 (SPSS, Chicago, IL, USA).

## Figures and Tables

**Figure 1 metabolites-12-00375-f001:**
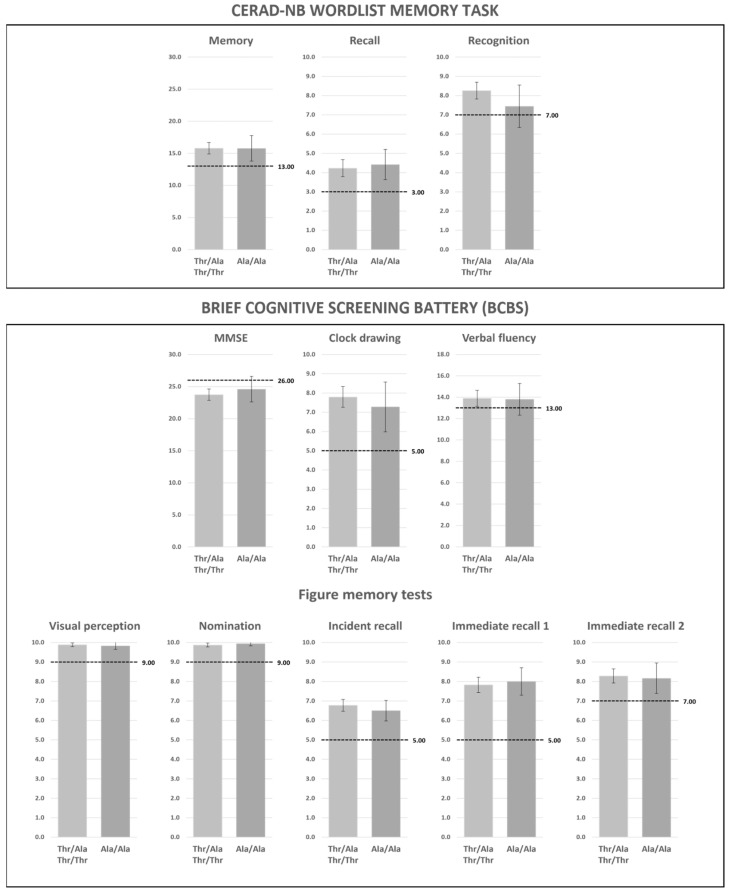
Cognitive tests results. Comparison between Ala/Ala and Thr/Ala-Thr/Thr genotypes following a recessive model. Scores are presented as mean; error bars indicate 95% confidence interval. Memory: *p* = 1.0. Recall: *p =* 0.7. Recognition: *p =* 0.1. Mini-Mental: *p =* 0.4. Clock drawing: *p =* 0.4. Verbal Fluency: *p =* 0.7. Visual perception: *p =* 0.6. Nomination: *p =* 0.5. Incident recall: *p =* 0.4. Immediate recall 1: *p =* 0.7. Immediate recall 2: *p =* 0.8. Dashed lines indicate cut-off values.

**Table 1 metabolites-12-00375-t001:** Clinical and laboratory characteristics of patients grouped according to Type 2 Deiodinase Thr92Ala genotype.

Baseline Characteristics	All (*n* = 100)	Ala/Ala (*n* = 18)	Thr/Ala-Thr/Thr (*n* = 82)	*p*
Age (years)	72 ± 6	73 ± 7	72 ± 6	0.56
Women-n (%)	61 (61)	10 (56)	51 (62)	0.61
Body Mass Index (Kg/m^2^)	30 ± 6	31 ± 5	30 ± 6	0.24
Diabetes-n (%)	51 (51)	9 (50)	42 (51)	0.92
Schooling level-n (%)				0.41
Never went to school	5 (5)	2 (11)	3 (4)	
Primary education	48 (48)	8 (44)	40 (49)	
Lower Secondary	29 (29)	6 (33)	23 (28)	
Upper Secondary	16 (16)	2 (11)	14 (17)	
Tertiary education	1 (1)	0 (0)	1 (1)	
Income-n (%)				0.09
Less than 1 MW	2 (2)	0 (0)	2 (2)	
One to less than two MWs	38 (38)	4 (22)	34 (41)	
Two to less than five MWs	55 (55)	14 (78)	41 (50)	
Over five MWs	3 (3)	0 (0)	3 (4)	
Laboratory				
TSH (mU/L)	1.83 (1.29–2.60)	2.23 (1.57–2.66)	1.78 (1.25–2.56)	0.44
Free T4 (ng/dL)	1.17 (1.06–1.29)	1.22 (1.07–1.29)	1.17 (1.05–1.29)	0.40

Comparison between Ala/Ala and Thr/Ala-Thr/Thr genotypes following a recessive model. Data are presented as number of patients (percentages), mean ± standard deviation, or median (interquartile range). The reference ranges for laboratory values are TSH 0.35–4.94 mU/L and free T4 0.70–1.48 ng/dL. MW: minimum wage.

**Table 2 metabolites-12-00375-t002:** Summary of the previous studies analyzing the association of the Thr92Ala DIO2 polymorphism and cognition.

Author	Country	Population Studied	*n*	Age (Years)	Ala allele Frequency	Cognitive Tests	Main Study Findings/Conclusions
Marcondes et al. [7]	Brazil	ASD patients	132	<18	47%	Autism behavior checklist, Vineland Adaptative Behaviour Scales II, non-verbal intelligence test SON-R 2½-7, SON-R 6-40, Weschler scale for intelligence, and autism treatment evaluation checklist	The frequency of the minor allele Ala92 DIO2 was no different in 132 ASD patients under the age of 18 (47%) compared with a local reference population (51%), neither correlated with ASD severity.
Luo et al. [8]	China	MCI patients	260	64 ± 6	MCI 47% Controls 45%	DSM-IV criteria to evaluate MCI	129 MCI patients were matched with 131 control subjects in a case-control design in which the Ala92 DIO2 allele frequency in the MCI group was no different than that of the control group.
Taylor et al. [9]	United Kingdom	Children from the Avon Longitudinal Study of Parents and Children birth cohort	3127	7 and 8	36.7%	“Extensive cognitive tests” (specific tests not mentioned in the published abstract)	116 (3.17%) children who had free T4 in the lowest quartile and the Thr92Ala substitution were more likely to have a total IQ score below 85 than individuals with free T4 above the lowest quartile without the substitution (odds ratio 3.03, 95% CI 1.38–6.67; *p* = 0.006. However, children with free T4 in the lowest quartile only were not associated with lower IQ, suggesting that thyroid hormone concentrations alone do not entirely reflect thyroid status.
Panicker et al. [15]	United Kingdom	Patients on a stable dose of T4 therapy from 28 primary care practices	552	Treatment (T4/T3) 57 ± 11 Control (T4 only) 58 ± 10	40%	General Health Questionnaire, 12-question version (GHQ-12), disease-specific thyroid symptom questionnaire (TSQ), Hospital Anxiety and Depression Scale questionnaire (HAD)	16% of patients on thyroid hormone replacement with the Ala92Ala genotype predicted both poorer psychological well-being on T4 monotherapy and improved response to combination T4/T3 assessed by the General Health Questionnaire 12 (GHQ-12).
Guo et al. [16]	China	MR patients	543	10 ± 3	38%	Chinese Wechsler Young Children Scale of Intelligence (C-WYCSI) Chinese Wechsler Intelligence Scale for Children (C-WISC)	No significant difference was found in the allele Ala92 DIO2 frequency among children with MR and controls (*p =* 0.5) in iodine-deficient areas of China.
Appelhof et al. [17]	The Netherlands	Patients with autoimmune primary hypothyroidism on T4 replacement	141	Wild type 47 ± 9; Heterozygous 48 ± 11; homozygous 52 ± 8	40%	Cognitive speed (Digit symbol, Memory Comparison Test paper-pencil version and computer version), Attention (Paced Auditory Serial Attention Task) and Memory (Digit symbol, digit span, California Verbal Learning Test and Rivermead)	The group of patients with theAla92Ala genotype had generally the worst scores on neurocognition, being significant in only 1 subtest of 21 tests applied.
Wouters et al. [18]	The Netherlands	Subjects from the Lifelines Cohort study	12,625	General population 48 ± 11; LT4 users 53 ± 12	33%	Ruff Figural Fluency Test (RFFT)	Thr92Ala polymorphism was not associated with cognitive functioning, neither in the general population nor in subjects on thyroid hormone replacement therapy or matched controls.

Abbreviations: ASD: Autism Spectrum Disorder; MCI: Mild Cognitive Impairment; MR: Mental Retardation; SON-R: Snijders-Oomen Nonverbal-Revised; DSM-IV: Diagnostic and statistical manual of mental disorders, 4th edition; IQ: Intelligence Quotient.

## Data Availability

Not applicable.

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
