# Peer review of "Type 2 Deiodinase Thr92Ala Polymorphism Is Not Associated with Cognitive Impairment in Older Adults: A Cross-Sectional Study"

_metabolites, 2022, doi:10.3390/metabo12050375_

Round 1
Reviewer 1 Report
The manuscript of Schwengber et al. entitled “Type 2 Deiodinase Thr92Ala polymorphism is not associated with cognitive impairment in older adults: a cross-sectional study” investigates the whether a common polymorphism (Thr92Ala) of the DIO2 enzyme is related to lessened cognitive function in older (> 65 years) adults. To achieve this goal, Authors recruited 100 outpatients from a tertiary care university hospital. The patients had euthyroid function and no previously known cognitive impairment. Exclusion criteria were well defined to eliminate the confounding effects of other medical conditions or therapeutics. The investigators performed diverse cognitive tests, verified the thyroid status by measuring serum free T4 and TSH levels and collected demographic data (educational level and economical data).
The study is well designed, the data are analyzed in an appropriate manner and presented in a comprehensive way in one Table and in one multipanel figure. Authors also describe the statistical data in the text.
The topic of the manuscript fits the scope of the special edition of “Metabolites”.
The manuscript is written in a clear fashion, and the results presented support the conclusion of the manuscript. The conclusion is “negative data” that has clear implications for future studies of this research field. Authors also provide reasonable explications for the lack of correlation reported.
This reviewer has only two minor comments to improve the manuscript.
- “Metabolites” has a wide-scope readership and not just endocrinologists: it would be helpful to mention the allele prevalence of Thr/Thr, Ala/Thr and Ala/Ala identified in the general population in other studies, with specific interest for the prevalence in Brazil. This should be included in the Introduction part.
- It would also greatly improve the quality of the manuscript if Authors could present an additional Table about the controversial studies mentioned on Page 2, Line 48, “The association between Thr92Ala DIO2 polymorphism and cognitive impairment in 48 clinical studies is controversial [7-9, 15-18]”. Authors should list the number of patients, age, cognitive tests used, Dio2 allele frequencies, and the data supporting the conclusions of those studies along with any other data that Authors deem useful to demonstrate the differences that might explain the controversial results obtained.Such a summary would also help to put the findings of the current study in perspective.
Reviewer 2 Report
The subjects should be better described - there is no mention of BMI or presence of IFG/DM, which are known to - eventually - have an impact on cognition.
Was a correction necessary for statistical analysis? [taking into account the number of comparisons between groups]
Finally a concern about the rationale of the study: if the work by Wouters et al in 2017 found no effect of the examined polymorphism in thousands of subjects how can a study of 100 subjects be expected to find any?
Round 2
Reviewer 2 Report
No further comments - the authors have delt adequately with this reviewer's comments.